# Contribution of Tin to the Strain Hardening of Self-Lubricating Sintered Al-30Sn Alloy and Its Wear Resistance under Dry Friction

**DOI:** 10.3390/ma16041356

**Published:** 2023-02-05

**Authors:** Nikolay M. Rusin, Alexander L. Skorentsev, Andrey I. Dmitriev

**Affiliations:** Institute of Strength Physics and Materials Science of Siberian Branch Russian Academy of Sciences (ISPMS SB RAS), 2/4, pr. Akademicheskii, 634055 Tomsk, Russia

**Keywords:** aluminum matrix alloy, equal channel angular pressing, structure, strength, dry friction, wear resistance

## Abstract

Aluminum alloys, which have been widely used in various manufacturing industries as an upper layer of bearing inserts, are alloyed with Sn to decrease the intensity of adhesive wear. A relationship between the mechanical properties, wear resistance, and structure of sintered Al-30Sn alloy containing a large amount of the soft phase was studied in this work. The above-mentioned characteristics were determined by testing the investigated material under compression and wear under dry friction in the pin-on-disk geometry at a sliding speed of 0.6 m/s and pressures of 1–5 MPa. The studied alloy was prepared by sintering of compacts consisting of a mixture of commercial powders in a vacuum furnace at a temperature of 600 °C for an hour. Then, the sintered Al-30Sn samples were subjected to processing by equal channel angular pressing (*ECAP*) with routes A and C. It has been established that the hardening value of the alloy subjected to *ECAP* virtually does not depend on the Sn content, but it depends on the number of passes and the processing route. The maximum increase in the strength of the alloy was found after the first and second passes. At the fixed Sn content, its effect on the wear resistance of the alloy does not depend on the strain hardening value of the aluminum matrix.

## 1. Introduction

Aluminum-based alloys have been widely used in various manufacturing industries due to their high specific strength, good thermal conductivity, corrosion resistance, and fatigue strength [1]. However, they tend to seizure with the surface of a hard counterbody under dry and boundary friction, even at low pressures. To decrease the intensity of adhesive wear of Al-based alloys, they are alloyed with soft metals (Sn or Pb), which do not dissolve in the aluminum matrix. Inclusions of these metals can serve as a solid lubricant for the friction surface [2,3,4,5,6,7]. With an increase in their content, the pressure of severe seizure of aluminum with steel increases, and the friction coefficient (µ) decreases to about 0.2 for the AO20-1 alloy [8,9]. The use of Sn as an antifriction additive made it possible to develop a number of commercial Al-Sn alloys (Standard 14113-78). These alloys are suitable for manufacture of bearing inserts operating at a pressure (*P*) up to 30 MPa and a sliding speed (*V*) up to 20 m/s. However, such alloys have poor wear resistance under boundary and dry friction due to their low strength and hardness.

During the crystallization of Al-Sn melt, Sn atoms are transferred to the surface of growing Al grains in the form of thin interlayers, which weaken the cohesive bonding between the grains. When a material with such a structure is subjected to loading, it can be prone to the formation of macroscopic bands of localized deformation. This process is especially easy when the interlayers are combined into a continuous net. As a result, the ductility of the aluminum alloy with high Sn content is reduced. Therefore, such alloys are recommended to use in the form of a thin coating firmly fixed on a hard substrate [9,10,11].

In some cases, the continuous Sn net in the Al-Sn alloys can be decomposed into isolated inclusions by severe plastic deformation (*SPD*). In this case, the ductility of the alloys is improved with increasing temperature and can reach high values [12,13]. For example, the AO20-1 alloy subjected to rolling and annealing can be deformed by 60% and significantly hardened (σB is about 120 MPa). However, when the Sn content is more than 10 vol.% (>20 mass.%), the continuous Sn net often does not break, even during *SPD* of the alloys [14].

Sintering is also a method of preparing Al-Sn alloys. In this case, a continuous skeleton of aluminum powders is formed in raw Al-Sn compacts, and Sn particles are located between these powders. During the sintering, the Al powders remain solid, and the skeleton is preserved [15,16,17]. Despite the fact that tin is melted in this case, it is not able to decompose the skeleton because the melt does not penetrate into many contacts of the solid particles due to a large wetting angle. With an increase in the sintering temperature, the Al solubility in liquid Sn increases, and the value of the wetting angle decreases. However, by the time the melt spreads over the compact and forms a continuous net [15], the mechanical contacts between the aluminum powders transform into sintering necks. That is, the Al matrix in the sintered material remains continuous and retains its bearing capacity even when the Sn content is higher than that of industrial alloys (>20 mass.%). As a result, the wear resistance of the alloys with high Sn content is improved under dry friction conditions.

Al and Sn are mutually insoluble in the solid state and do not affect the mechanical properties of each other. That is, the strength of sintered Al-Sn alloys is an additive value and is determined by their volumetric phase ratio. Tin is not hardened during deformation, and the strength of the deformed two-phase Al-Sn alloys depends only on the strain hardening degree of the Al grains [14]. The strength of the alloys can be additionally increased by using a solid solution [18,19,20] or dispersion [21,22,23] hardening of the aluminum matrix. However, in this case, it is necessary to avoid the presence of the alloying elements in Sn and, as a consequence, the deterioration of its lubricating properties [24].

In this regard, strain hardening of the aluminum matrix by *SPD* of the sintered samples can be considered as a promising way to increase the strength and wear resistance of two-phase Al-Sn composites. Among the *SPD* methods, the processing of materials by equal channel angular pressing (*ECAP*) is of particular interest because dimensions of the deformed sample are not changed, even after significant deformations during *ECAP* [25,26]. At the same time, due to the choice of different *ECAP* routes, it is possible to change the size and shape of the structural elements of the processed material in a wide range. It is also known that the *ECAP* processing leads to a significant increase in the strength and wear resistance of different aluminum alloys [13,27,28]. These facts make it possible to prepare the Al-Sn alloys with a durable matrix and optimal two-phase macrostructure of the friction surface.

Therefore, the main purpose of this work was to determine the effect of severe *ECAP* processing on the mechanical properties and wear resistance of sintered Al-30Sn alloy containing a large amount of the soft phase.

## 2. Materials and Methods

The studied alloys were prepared by sintering of compacts consisting of a mixture of elemental commercial Al (ASD-4) and Sn (PO 2) powders. The powders were mixed in a ratio of 70/30 (mass.%), respectively, in a ball mixer and pressed into compacts 60 mm long with a cross-section of 10 × 10 mm for *ECAP* processing. The initial porosity of the compacts was about 10%. The compacts were sintered at a temperature of 600 °C for an hour in a vacuum furnace with a residual gas pressure of not more than 10^−2^ Pa. Then, they were subjected to *ECAP* processing in a steel die with channels 10 × 10 mm intersecting at a right angle (Figure 1a). The processing was carried out at 200 °C with route A (*ECAP-A*) without sample rotation between the passes and with route C (*ECAP-C*) with a turn of 180° after each pass [26,29]. During each pass of the *ECAP* processing, the deformation of the sample was carried out according to the scheme of simple shear with an intensity (γ) of about 2 [15,26].

Samples with the size of 5 × 5 × 10 mm were cut from the middle of the prepared briquettes for compression test using the Instron-1185 machine. The rate of compression was 0.5 mm/min. Wear tests were carried out according to pin-on-disk scheme (Figure 1b) (Standard ASTM G99-95a) under dry friction using the tribotester (Tribotechnic, France). The area of the friction surface of the alloy samples was 2 × 2 mm. The wear rate (*Ih*) of the alloys was defined as the ratio of reduction in the specimen height to the sliding distance (*L*). The counterbody (disk) with a diameter of 50 mm and a thickness of 10 mm was prepared from a hardened structural steel 40H (AISI 5140 steel) with a hardness of 48 ± 2 HRC. The sliding velocity (*V*) was 0.6 m/s, and the sliding distance was 1000 m for each test. The pressure on the friction surface (*P*) was 1–5 MPa. The sample height was measured with a micrometer with an accuracy of 0.01 mm. The standard deviation of the *Ih* value was ± 0.02 µm/m. The friction surfaces of the samples and counterbody were prepared by mechanical grinding with emery paper and subsequent polishing on a cloth with applied diamond paste having hard particles smaller than 1 μm. The surfaces were cleaned with acetone before the wear tests. The compression and wear tests were repeated three times under the same conditions.

The structure of the samples before and after the tests was studied using optical AXIOVERT-200MAT and scanning electron LEO EVO 50 (Carl Zeiss, Germany) microscopes that were provided by shared use center Nanotech of the Institute of Strength Physics and Materials Science SB RAS (ISPMS SB RAS, Tomsk, Russia). Metallographic cross-sections were prepared according to the above-mentioned method of surface preparation and subjected to subsequent short-term chemical etching in 4% solution of nitric acid in ethanol.

The microstructure studies by a method of transmission electron microscopy (TEM) were carried out using the TEM-125K microscope with a selector diaphragm diameter of 1 μm.

X-ray diffraction data were obtained using DRON-7 diffractometer with the use of Co*K*α radiation without a monochromator in symmetric reflection geometry. The recording was performed within a range of angles (2θ) 25° ≤ 2θ ≤ 165° with a step of 0.05°.

## 3. Results and Discussion

Evolution of the structure of the Al-30Sn alloy during its sintering and subsequent *ECAP* processing is shown in Figure 2. It can be seen that Sn powders are separated from each other by numerous aluminum powders in the raw compact (Figure 2a). During the sintering, tin is melted, spreads over the compact, and forms a net. The large size of aluminum grains (Figure 2b) indicates that when the Sn melt is saturated with aluminum, the stage of active recrystallization of solid phase particles through the melt takes place. It can be seen from the X-ray phase analysis data that no new phases and compounds are formed during the sintering (Figure 3), which is consistent with the Al-Sn phase diagram.

During the processing of sintered samples by *ECAP*, their macrostructure is significantly changed, which is especially noticeable in the flow plane of the material. Under the *ECAP-A* processing, the Al matrix grains, as well as Sn inclusions located between them, are elongated in the flow direction of the material (Figure 2c). The density of the Al-30Sn alloy processed by *ECAP* is close to the theoretical one (within the measurement error), while the porosity of the alloy after sintering is about 1%. At the same time, the initial large aluminum grains of the Al-30Sn alloy are fragmented into smaller subgrains (Figure 4). The aluminum matrix after sintering consists of large grains 70–150 µm in size. After the first *ECAP*, the initial grains in the matrix are fragmented into subgrains up to 3 μm in size (Figure 4a). With an increase in the number of *ECAP* passes to four, the subgrain structure of the matrix becomes more uniform. In the case of *ECAP-A*, the subgrain size is 0.5–1 μm (Figure 4b). As a result, the aluminum matrix is hardened due to the Hall–Petch effect. The thickness of the Al and Sn interlayers is continuously decreased with an increasing the number of *ECAP-A* passes.

In the case of route C, the direction of a simple shear in the deformation zone is reversed after each pass [26]. Therefore, the aluminum grains tend to return to their original shape during each even pass. A large interfacial distance allows more active growth of the matrix subgrains due to a dynamic recrystallization (Figure 4c). Thus, the hardening of the material during *ECAP* with route C is carried out more slowly than in the case of route A (Figure 5).

It also follows from the data in Figure 5 that the maximum increase in strain hardening of the samples, both with and without tin, takes place during the first two *ECAP* passes. The strength of the samples is slightly increased during the subsequent passes. However, in this case, the risk of disintegration of Sn interlayers and exhaustion of their plasticity resource increases due to their significant thinning. Moreover, with an increase in the number of *ECAP* passes with route A, the area of uniform phase distribution decreases due to the “end effect” that takes place during this processing [26]. Therefore, in this work, the wear properties of the materials were studied using samples subjected to two passes by *ECAP*. Their structure is shown in Figure 6.

It can be seen from the presented images that the macrostructure of the Al-30Sn alloy after two passes with route C remains coarse-grained (Figure 6b), while the structure of the material becomes layered in the material flow plane after two passes by *ECAP-A* (Figure 6a). If a flow plane of the material with such a layered structure is used as a friction surface, the distance between Sn inclusions which are sources of solid lubricant will be much smaller than that in the initial sintered material (Figure 2b) or sample after *ECAP* with route C. It was found in [30] that the processing of Al-Sn alloys by *ECAP-C* is less effective in increasing their wear resistance than *ECAP-A*. Perhaps the reason for this lies in the different distances between the Sn interlayers, which act as sources of solid lubricant and can be smeared on the friction surface in the sliding direction.

The fact that Sn can be extruded on the friction surface from the sample is confirmed by an image of the surface of the alloy processed by *ECAP-A*, which was made a day after preparation of the metallographic cross-section (Figure 7). It can be seen that even residual internal stresses can squeeze out a certain amount of tin on the cross-section surface. However, tin is extruded in the form of “fountains” from the middle of the wide Sn interlayers and not along their entire length, as one might expect. Tin is not extruded from the thin interlayers because it is held by friction forces and strong adhesive bonds with aluminum matrix grains. If the regularity of the Sn location places remains the same during the friction, it is difficult to achieve its uniform distribution over the friction surface after a significant thinning of the Sn interlayers.

The maximum increase in the wear resistance of Al-Sn composites with a high tin content is achieved after two passes by *ECAP* with both of the above-mentioned routes [30]. A further increase in the number of *ECAP* passes does not lead to an increase in the wear resistance of the composites and even decreases it, despite the fact that thinning of the phases’ interlayers continues in the case of route A [15]. Apparently, thinning of the Sn interlayers and, as a result, an increase in the area of the interfacial surface, which is harder than tin, do not allow increasing the volume of the solid lubricant extruded on the friction surface. In addition, a significant thinning of the Sn interlayers can lead to a sharp decrease in their plasticity resource or the formation of discontinuities at their boundary with the aluminum matrix, which can also affect the value of *Ih*. Examples of interfacial separations can be seen in Figure 2c and Figure 7.

In view of the above, the study of wear properties under dry friction of the sintered Al-30Sn alloy was carried out using sintered samples and samples subjected twice to *ECAP* with route A. For comparison, the wear rate values of the alloy subjected to *ECAP-C* are shown in Table 1. It can be seen that after the treatment with this route, the material has a lower wear resistance compared to the alloy processed with route A.

The test results of the sintered Al-30Sn alloy are shown in the top line in Table 1. It can be seen that at a sliding speed of 0.6 m/s and a pressure of 1 MPa, the wear resistance of the Al-30Sn alloy sample is lower than that of the pure aluminum one. The wear rate of the aluminum sample is increased with increasing pressure on the friction surface. The same trend is observed in the case of the Al-30Sn alloy, but the *Ih* value is increased more slowly in the presence of tin in the aluminum matrix, since its positive effect as a solid lubricant takes place. As a result, at *P* = 5 MPa, the *Ih* value of the Al-30Sn alloy becomes one third less than that of pure aluminum.

The strength of the sintered samples after the first two passes of *ECAP* is increased noticeably (Figure 5), and this fact contributes to the improvement of their wear resistance. However, the nature of dependence of the *Ih* value on the applied pressure remains the same as before the *ECAP* processing of the samples. In the case of sintered samples subjected to subsequent *ECAP*, the wear resistance of sintered aluminum is also higher than that of the Al-30 alloy at *P* = 1 MPa because the decrease in the *Ih* value caused by hardening in both cases was about 20%. With increasing pressure on the friction surface, the effect of matrix hardening on the *Ih* value was the same in both materials regardless of their Sn content. As a result, due to the effect of Sn, the processed Al-30Sn alloy samples have higher wear resistance than the processed pure aluminum ones.

It should be noted that at each applied pressure, the difference in *Ih* values (*ΔIh*) between the samples of Al-30Sn alloy and pure aluminum remains the same, both in their sintered and hardened states. For example, at *P* = 1 MPa, the value of *ΔIh* is − 0.02 μm/m both for the sintered samples and the samples subjected to subsequent *ECAP-A*. At *P* = 3 MPa, the above-mentioned difference *ΔIh* is + 0.16 µm/m, and *ΔIh* = + 0.20 µm/m in the case of *P* = 5 MPa. That is, at a fixed amount of tin, its effect on the wear resistance of the aluminum matrix is also fixed. At low pressures, this effect is negative, and it is positive at elevated pressures. This regularity is also preserved after two passes of the samples by the *ECAP* method. Therefore, a fixed amount of tin improves the wear resistance of the aluminum matrix by a fixed value, which does not depend on the grain structure of the matrix and features of the Sn inclusions’ distribution. This indicates that *Ih* is at least a two-term value, one of the terms of which is determined by Sn content in the aluminum matrix.

It can be seen from images of the subsurface structure shown in Figure 8 that the wear mechanism of the investigated alloy is not changed after the *ECAP* processing despite the high strain hardening of the matrix, refinement of the aluminum grains, and thinning of the Sn interlayers. The main wear mechanism of the sintered Al-30Sn alloy under the dry friction conditions is a fracture of the upper brittle mechanically mixed layer and a delamination of highly deformed matrix grains along Sn interlayers. Large cracks (shown by arrows in Figure 8) are formed as a result of deformation localization, thinning of the Sn interlayers, and exhaustion of their plasticity resource.

The second term of the *ΔIh* value is determined by strength of the aluminum matrix. Its contribution to the wear resistance of the Al-30Sn alloy can also be estimated from the analysis of the *ΔIh* values obtained at different pressures on the friction surface by comparing the *Ih* values for sintered samples and samples processed by *ECAP*. With an increase in pressure up to 3 MPa, the value of *ΔIh* is – 0.10 µm/m, and *ΔIh* = – 0.14 µm/m at *P* = 5 MPa both for pure aluminum and Al-30Sn alloy samples (Table 1). It follows from the obtained data that, as a result of the *ECAP* processing, an increase in the wear resistance of pure aluminum samples is the same as that of the Al-30Sn alloy ones. It can be argued that: (a) contribution of the matrix hardening of the aluminum alloys to their wear resistance does not depend on the presence of a soft Sn phase in them, but it is determined by the strain hardening value; (b) strain hardening of the aluminum matrix does not depend on the presence of tin in the alloys, but it is determined only by the deformation value of them; (c) tin in the aluminum matrix is not hardened either during *ECAP* or under wear tests. That is, the strength of the Al-Sn system alloys is an additive value, and the contribution of the Sn phase to its value is determined by the Sn content regardless of the nature of its distribution in the aluminum matrix.

From a practical point of view, it should be noted that the Sn content will have a positive effect on the wear resistance of Al-Sn alloys at high operating pressures in friction units.

## 4. Conclusions

1. The processing of the Al-30Sn alloy by equal channel angular pressing leads to its significant hardening, especially during the first two passes. During the subsequent passes, the strain hardening rate of the material is significantly reduced, and its plasticity decreases due to the fracture of the intergranular Sn interlayers.

2. *ECAP* with route A transforms the two-phase structure of the Al-30Sn alloy into a layered one in the flow plane of the material, and the thickness of the interlayers decreases with an increase in the number of passes. In the case of *ECAP* with route C, the aluminum grains tend to return to their original shape at each even pass.

3. Strain hardening of the aluminum matrix in the Al-30Sn alloy subjected to *ECAP* leads to an increase in its wear resistance under dry friction against steel. This effect is determined by the degree of the matrix hardening and does not depend on the Sn distribution.

4. The effect of Sn on the wear resistance of the aluminum matrix under dry friction conditions depends on the applied pressure. At low pressures, this effect is weak and negative, but it becomes positive and rises with increasing the pressure. Moreover, this effect depends only on the Sn content and is independent on the strength of the aluminum matrix.

## Figures and Tables

**Figure 1 materials-16-01356-f001:**
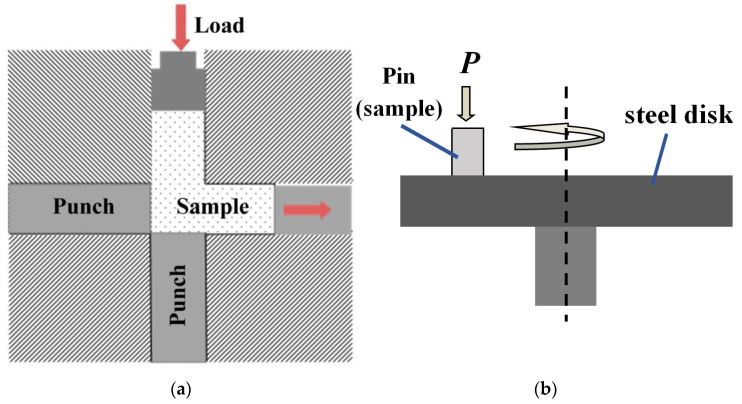
Sample processing scheme in the die for *ECAP* (**a**) and scheme pin-on-disk of the wear tests (**b**).

**Figure 2 materials-16-01356-f002:**
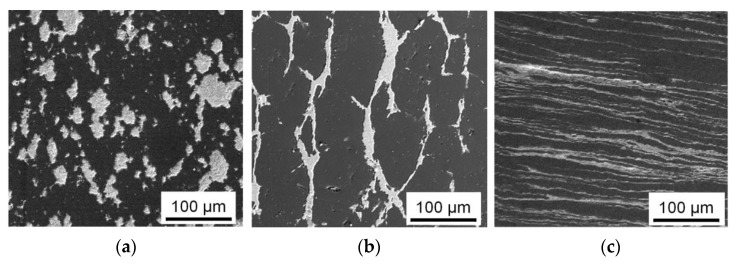
SEM (SE) images of the structure of green (**a**), sintered (**b**), and subjected to four *ECAP-A* passes (**c**) Al-30Sn alloy.

**Figure 3 materials-16-01356-f003:**
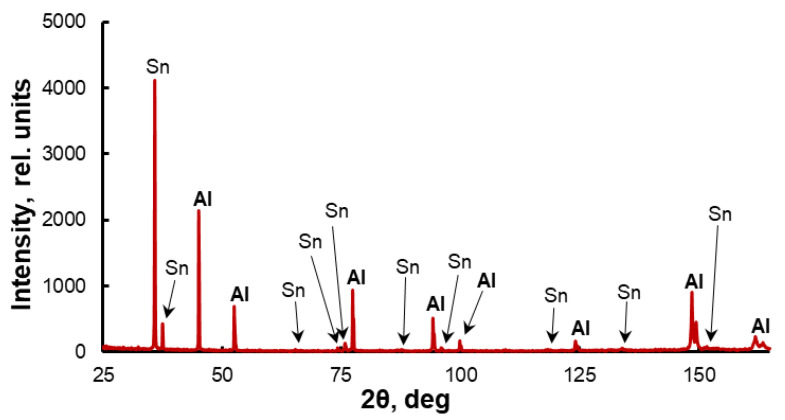
X-ray diffraction pattern of sintered Al-30Sn alloy.

**Figure 4 materials-16-01356-f004:**
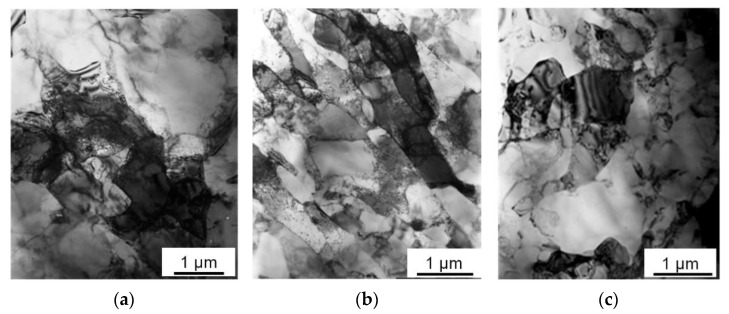
Bright field TEM images of an aluminum matrix in the Al–30Sn alloy after different number of *ECAP* passes: (**a**) 1 *ECAP*; (**b**) 4 *ECAP-A*; (**c**) 4 *ECAP-C*.

**Figure 5 materials-16-01356-f005:**
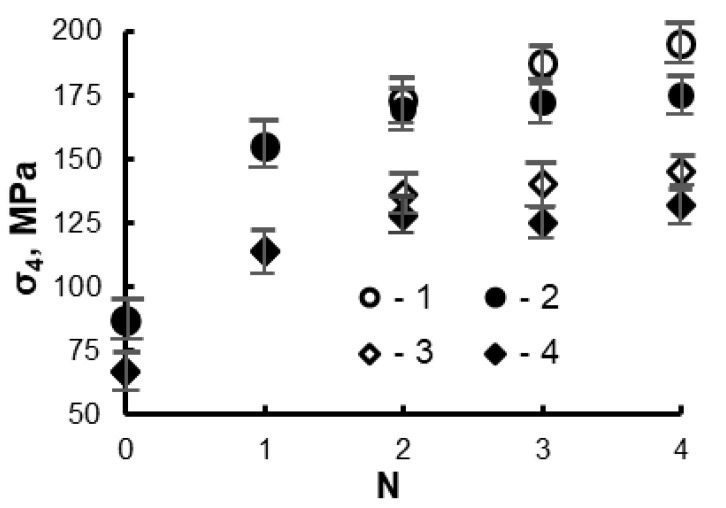
Effect of the number of *ECAP* passes (*N*) with routes A (1, 3) and C (2, 4) on the flow stress of the samples after their compression by 4% (σ_4_). 1, 2—sintered pure aluminum; 3, 4—Al-30Sn alloy.

**Figure 6 materials-16-01356-f006:**
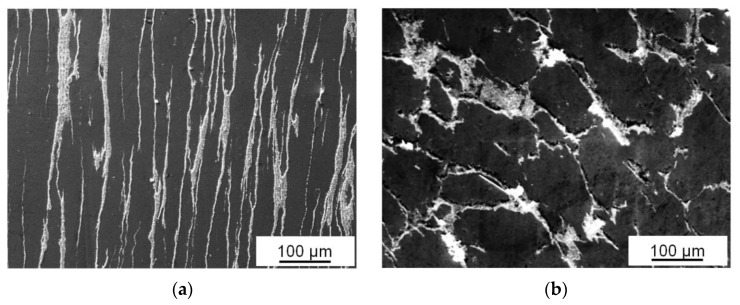
SEM (SE) images of the structure of Al-30Sn alloy in the flow plane of the sample after two passes by *ECAP* with routes A (**a**) and C (**b**).

**Figure 7 materials-16-01356-f007:**
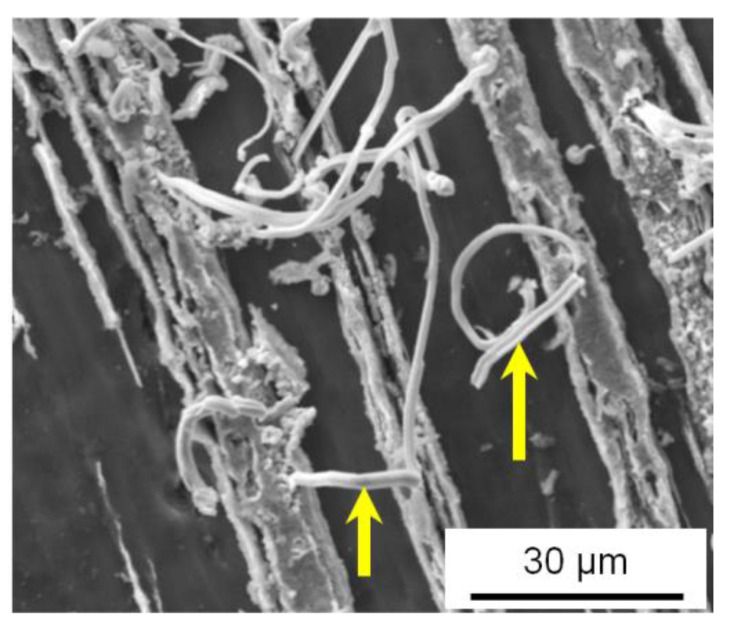
SEM (SE) image of the cross-section surface of Al-30Sn alloy subjected to three passes by *ECAP-A*. Sn extruded by internal stresses is shown by arrows.

**Figure 8 materials-16-01356-f008:**
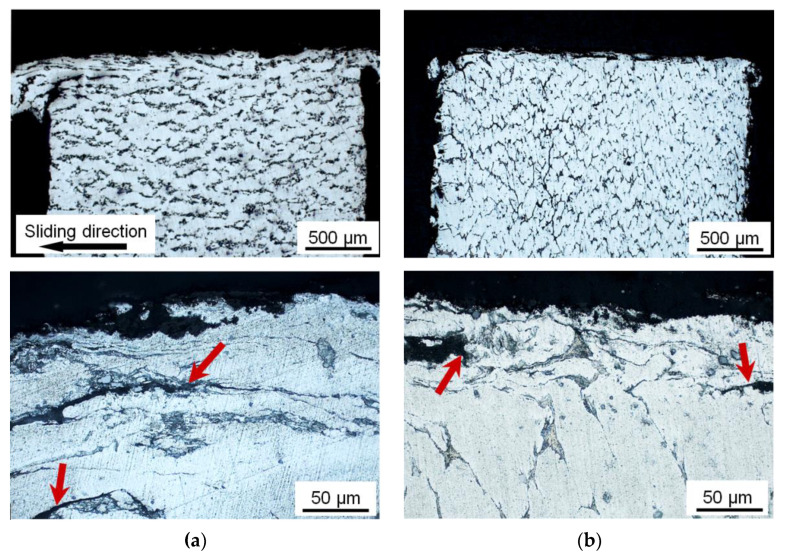
Structure of the subsurface layer of the sintered Al-30Sn alloy after dry friction against steel. Number of *ECAP-A* passes: (**a**) 0 (sintered); (**b**) 2. *P* = 5 MPa. Locations of crack propagation are shown by arrows.

**Table 1 materials-16-01356-t001:** Effect of pressure (*P*) and number of *ECAP* passes (*N*) with routes A and C on the wear rate (*Ih*) of the Al-30Sn alloy and sintered pure aluminum under dry friction against steel. *V* = 0.6 m/s, *L* = 500 m.

Composition	Number of *ECAP* Passes, *N*	Wear Rate *Ih*, µm/m		
		1 MPa	3 MPa	5 MPa
Al-30Sn	0	0.12	0.30	0.42
	2A 2C	0.10 0.11	0.20 0.28	0.28 0.36
Al	0	0.10	0.46	0.62
	2A	0.08	0.36	0.48

## Data Availability

Data sharing is not applicable.

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
