# Peer review of "Contribution of Tin to the Strain Hardening of Self-Lubricating Sintered Al-30Sn Alloy and Its Wear Resistance under Dry Friction"

_materials, 2023, doi:10.3390/ma16041356_

Round 1

Reviewer 1 Report

Dear Editor 

Article entitled " Contribution of tin to the strain hardening of self-lubricating sintered Al-30Sn alloy and its wear resistance under dry friction" , author stated his research work in well manner. Hence accept the article in the present format for publication.  

Author Response

Dear reviewer, thank you for your feedback on the manuscript. The quality of the English language of the manuscript has been improved.

Reviewer 2 Report

The paper deals with the dry friction wear resistance of Al-30Sn alloy. The resistance is measured by the pin-on-disk method at a defined velocity while varying the pressure of the specimen and the steel friction element. The method chosen by the authors for the research is described in ASTM G99-95a. However, this standard is not cited anywhere in the text, nor is it listed among the references.

The strengths of the paper are the clearly done research and the overview of by whom and what has been achieved in the issue discussed. In my opinion, the strength is the application of the chosen methods in the paper and thus the results obtained. These results then provide a wealth of interesting information from the experiments performed and their interpretations. The thesis is clearly written. The role of tin in the alloy is explained in the thesis. The influence of tin in terms of just wear resistance, its distribution in the alloy and what role it plays during loading is discussed in a clear and lucid manner. I did not find any formal errors or typos or deficiencies in marking in the paper.

On the other hand, however, I find the experiment and its evaluation lacking, and this is my personal opinion, the display and discussion of even common tribological parameters. The normal and frictional force waveforms are normally recorded during the measurements. From these, most experimental equipment determines the coefficient of friction. It is its gradient, whether positive or negative, that gives a more comprehensive understanding of the tribological processes occurring during the test. This is also with regard to wear resistance. This is only a recommendation.

Otherwise, the work is complex. Once again I would like to highlight the discussion of the role of tin in the alloy, which in my opinion is very detailed, clear and explains clearly what happens under load. The opinions are appropriately supplemented with TEM images.

In conclusion, I positively evaluate the overall summary in two points. However, the practical application of the results is also worth mentioning. Specifically, where the results can be applied in industry, practice, manufacturing or even further research and development.

Author Response

 Dear reviewer, thank you for your feedback on the manuscript.

[The method chosen by the authors for the research is described in ASTM G99-95a. However, this standard is not cited anywhere in the text, nor is it listed among the references.]

A reference to this standard has been added to the text of the manuscript.

[In conclusion, I positively evaluate the overall summary in two points. However, the practical application of the results is also worth mentioning. Specifically, where the results can be applied in industry, practice, manufacturing or even further research and development.]

The practical application of the results of the work has been added to the text of the manuscript.

Reviewer 3 Report

Editor, materials

Title: “Contribution of tin to the strain hardening of self-lubricating sintered Al-30Sn alloy and its wear resistance under dry friction”

.

Manuscript Number: materials - 2186963

Dear Editor,

        I am attaching my review comments of the manuscript on a paper entitled “Contribution of tin to the strain hardening of self-lubricating sintered Al-30Sn alloy and its wear resistance under dry friction”.

In this paper, the authors have studied the influence of severe plastic deformation SPD using ECAP on the microstructure, compression strength, and wear of Al-30Sn sintered samples. The ECAP improves compression strength and wear resistance. The study is interesting and focused on improving the properties of the Al-30Sn t alloy. However, different points of shortage can be noted in the paper. The reviewer suggests accepting this paper for publication in the materials after a major revision to cover the following comments. 

1-    Please rewrite the abstract; it must be comprehensive and contain some details about aims, experimental work, results, and correct conclusions.

2-   English needs to be corrected throughout the manuscript. The authors need deep revision, as many sentences are strange, and the meaning cannot be reached easily.  

3-      Please use the general expressions used in the field of SPD and wear, like grain size, not structural elements, ECAP die, not ECAP mold, applied load, not pressure, and others.

4-    The materials and methods part needs to be supported by the figures of the ECAP die, ECAP samples, and wear test.

5-      The total imposed strain value on the final bar must be added to the paper. That includes the die angles and corner radius.

6-    Why did the ECAP processing stop just after four passes? If for samples fracture occurs, please confirm with photos.

7-    The authors need to measure the density, especially after the ECAP.

8-    he authors measure the wear intensity Ih. Interestingly it is a new parameter in measuring wear resistance. However, if authors insist on using it, they need to provide more details about the size and dimensions of steel disk use. Moreover, how the change in the sample height (height or thickness decrease during the wear test measured) and how it was considered homogenized decrease across the surface of the sample (how the decrease considers the constant across the sample surface).

9-    Why do no phases not appear in figure 2?

10-  The grain size calculations must be added before and after the ECAP processing.

11-  Please rewrite the figure 3 caption.

12-  Microstructure part is too weak. It must be enhanced with more discussion

13-  Figure 4 must be enhanced; it is complicated to follow the results.

14-   The author must measure the hardness or microhardness. Moreover, the hardness or microhardness must be related to the grain size and the wear results.

15-  The author must enhance the discussion of the results. The discussion needs to be stronger and supported by previous works for the same alloy.

16-   Please reconstruct conclusions. It is a general conclusion without a deep concentration on the obtained results. 

Author Response

Dear reviewer, thank you for your feedback on the manuscript.

  • [Please rewrite the abstract; it must be comprehensive and contain some details about aims, experimental work, results, and correct conclusions.]

The Abstract section has been substantially revised.

2-   [English needs to be corrected throughout the manuscript. The authors need deep revision, as many sentences are strange, and the meaning cannot be reached easily.]

The quality of the English language of the manuscript has been improved.  

3-      [Please use the general expressions used in the field of SPD and wear, like grain size, not structural elements, ECAP die, not ECAP mold, applied load, not pressure, and others.]

We agree with several reviewer's comments about the terms. For example, term “ECAP mold” has been replaced by “ECAP die”. However, some of the terms cited by the reviewer were spelled correctly in the manuscript.  For example, we use pressure, but not applied load, because the pressure is defined as the applied load divided by the area of the tested material.

Term “grain size” does not always correspond to term “structural elements”. It is written in the manuscript: “…due to the choice of different ECAP routes, it is possible to change the size and shape of the structural elements of the processed sample in a wide range.” For example, tin interlayers are present as structural elements in the investigated material, their size and shape are changed significantly under the ECAP with route A processing, in contrast to the size of tin grains, which is not capable of grain refinement and deformation hardening due to its high homological temperature.

4-   [ The materials and methods part needs to be supported by the figures of the ECAP die, ECAP samples, and wear test.]

  The information about processing in the ECAP die and scheme of the wear tests has been added.

Sample sizes are specified in Section 2: “The powders were mixed in a ratio of 70/30 (mass. %) respectively in a ball mixer and pressed into compacts 60 mm long with a cross section of 10x10 mm for ECAP processing.”

5-      [The total imposed strain value on the final bar must be added to the paper. That includes the die angles and corner radius.]

The scheme of processing in the ECAP die has been added in Figure 1. The die angles and corner radius can be seen in this Figure. During the ECAP processing, the deformation of the sample is carried out according to the scheme of simple shear with the intensity of about 2. This information has been added to the manuscript.

6-    [Why did the ECAP processing stop just after four passes? If for samples fracture occurs, please confirm with photos.]

ECAP processing was carried out up to four passes, since with an increase in the number of ECAP passes, the risk of decomposition of Sn interlayers and exhaustion of their plasticity resource is increased due to their strong thinning. It is written in the manuscript: “It also follows from the data in Figure 4 that the maximum increase in strain hardening of the samples, both with and without tin, takes place during the first two ECAP passes. The strength of the samples is slightly increased during the subsequent passes. However, in this case, the risk of disintegration of tin interlayers and exhaustion of the plasticity resource is increased due to their strong thinning.”

The second reason is that with an increase in the number of ECAP passes with route A, the area of uniform phase distribution decreases due to the “end effect” that takes place during the processing. This information has been added to the manuscript.

7- [The authors need to measure the density, especially after the ECAP.]

This information has been added to the manuscript.

8-   [The authors measure the wear intensity Ih. Interestingly it is a new parameter in measuring wear resistance. However, if authors insist on using it, they need to provide more details about the size and dimensions of steel disk use. Moreover, how the change in the sample height (height or thickness decrease during the wear test measured) and how it was considered homogenized decrease across the surface of the sample (how the decrease considers the constant across the sample surface).]

Wear intensity (wear rate) Ih is a well-known characteristic which is defined as the ratio of the amount of sample wear to the friction path. The amount of wear is defined as a decrease in the mass of the sample. The change in mass is calculated as the density of the sample multiplied by the change in its volume. The density of the sample, as well as its cross-sectional area, remain unchanged during the pin-on disk test, and only its height changes. Consequently, the amount of sample wear can be measured as a change in its mass or height. In the case of aluminum samples, it is not entirely correct to measure the wear of the sample by changing the mass, since part of the worn material strongly adheres to the sample.

Term “wear intensity” has been replaced by “wear rate”.

The dimensions of the disk (counterbody) corresponded to the standard ASTM G99-95a. The sample height was measured with a micrometer with an accuracy of 0.01 mm. This information has been added to the manuscript.

9-    [Why do no phases not appear in figure 2?]

It is written in the manuscript: “It can be seen from the X-ray phase analysis data that no new phases and compounds are formed during the sintering (Figure 2), which is consistent with the Al-Sn phase diagram”. Thus, only 2 phases Al and Sn are present in the X-ray diffraction pattern. This also follows from the two-phase Al-Sn diagram, the elements of which do not form compounds with each other and do not dissolve in each other in the solid state.

10-  [The grain size calculations must be added before and after the ECAP processing.]

 This information has been added to the manuscript.

11-  [Please rewrite the figure 3 caption.]

The figure 3 caption has been corrected.

12-  [Microstructure part is too weak. It must be enhanced with more discussion.]

This part of the text has been corrected.

13-  [Figure 4 must be enhanced; it is complicated to follow the results.]

Figure 4 has been improved.

14-   [The author must measure the hardness or microhardness. Moreover, the hardness or microhardness must be related to the grain size and the wear results.]

It is not correct to measure the microhardness of the investigated materials because with an increase in the number of ECAP with route A, aluminum grains and tin interlayers become very thin, and it is almost impossible to find areas with a pure aluminum matrix. Brinell hardness measurements were taken, and its dependence on the number of ECAP passes correlates well with the strength. During the operation of the friction unit, the aluminum alloy bearing experiences loads similar to the compression test process. Therefore, a compression test was chosen as a method for measuring the mechanical properties of the Al-30Sn alloy.

15-  [The author must enhance the discussion of the results. The discussion needs to be stronger and supported by previous works for the same alloy.]

This part of the text has been substantially corrected.

16-   [Please reconstruct conclusions. It is a general conclusion without a deep concentration on the obtained results.]

The Conclusions section has been substantially revised.

Round 2

Reviewer 3 Report

I am pleased to  Accept the paper in present form.

Best Regards